# Recurrent SARS-CoV-2 Serology Testing and Pandemic Anxiety: A Study of Pediatric Healthcare Workers

**DOI:** 10.3390/ijerph19159562

**Published:** 2022-08-03

**Authors:** Natasha Li, Sarah R. Martin, Theodore W. Heyming, Chloe Knudsen-Robbins, Terence Sanger, Zeev N. Kain

**Affiliations:** 1Children’s Health of Orange County, Orange, CA 92868, USA; nli@emsoc.net (N.L.); sarahm7@hs.uci.edu (S.R.M.); tsanger@choc.org (T.S.); zkain@uci.edu (Z.N.K.); 2Department of Anesthesiology and Perioperative Care, University of California, Irvine, CA 92868, USA; 3Center on Stress & Health, University of California, Irvine, CA 92868, USA; 4Department of Emergency Medicine, University of California, Irvine, CA 92868, USA; 5School of Medicine, University of Pittsburgh, Pittsburgh, PA 15213, USA; chloekr@g.ucla.edu; 6Department of Electrical Engineering and Computer Science, University of California, Irvine, CA 92697, USA

**Keywords:** anxiety, COVID-19, pediatrics, antibody testing

## Abstract

Background: Limited access to SARS-CoV-2 testing has been identified as a potential source of anxiety among healthcare workers (HCWs), but the impact of repeated testing on pandemic-related anxiety in pediatric HCWs has not been examined. We sought to understand the impact of repeated SARS-CoV-2 antibody testing on pediatric HCWs’ COVID-19 anxiety. Methods: This longitudinal cohort study was conducted between April and July 2020. Participants, 362 pediatric HCWs, underwent rapid SARS-CoV-2 antibody testing either every 96 h or weekly and were asked to rate their COVID-19 anxiety on a visual analog scale. Changes in self-reported anxiety from the study baseline were calculated for each testing day response. Bivariate analyses, repeated measures, and logistic regression analyses were performed to examine demographics associated with changes in anxiety. Results: Baseline COVID-19 anxiety was significantly higher in HCWs with less than 10 years of experience (*Z* = −2.63, *p* = 0.009), in females compared to males (*Z* = −3.66 *p* < 0.001), and in nurses compared to other HCWs (*F* (3,302) = 6.04, *p* = 0.003). After excluding participants who received a positive test result, repeated measures analyses indicated that anxiety decreased over time (*F* (5,835) = 3.14, *p* = 0.008). Of the HCWs who reported decreased anxiety, 57 (29.8%) had a clinically meaningful decrease (≥30%) and Emergency Department (ED) HCWs were 1.97 times more likely to report a clinically meaningful decrease in anxiety (*X*^2^ (1) = 5.05, *p* = 0.025). Conclusions: The results suggest that repeated SARS-CoV-2 antibody serology testing is associated with decreased COVID-19 anxiety in HCWs. Routine screening for the disease may be a helpful strategy in attenuating pandemic-related anxiety in pediatric HCWs.

## 1. Introduction

The COVID-19 pandemic has ravaged the physical and mental health of millions and severely strained health system infrastructures [1,2,3]. In July 2020, more than half of US adults reported that worry and stress related to the pandemic were negatively affecting their mental health, and compared to 2018, adults in 2020 were eight times more likely to fit criteria for serious mental distress [4,5]. Healthcare workers (HCWs) in particular face unique risks and stressors, including an increased likelihood of exposure to COVID-19 and fear of infecting one’s family and social circle, as well as increased workload and potentially unfamiliar work-related duties [6,7,8]. In a systematic review of 14 studies regarding post-traumatic stress, d’Ettore et al. found rates of post-traumatic stress symptoms in HCWs during the pandemic that ranged from 2.1% to as high as 73.4% [9]. In light of the continued pandemic and the need to nurture a healthy healthcare workforce, it is vitally important to identify protective factors and interventions regarding HCWs’ mental health.

Access to SARS-CoV-2 testing has been identified as a potential source of anxiety as well as an opportunity for possible intervention, not only for HCWs but also for the general public [10,11,12]. Existing data on the impact of screening for disease/risk for disease and anxiety is somewhat equivocal, an unsurprising finding given the variety of diseases studied and the availability and success of potential treatments/preventions/interventions [13,14,15,16,17]. Traditional tenets of screening include the possibility of a treatment or intervention altering the course of the disease if disease status is known, the practical and economic feasibility of widespread testing, the positive evaluation of risk/benefit, the profile of false positive/negative tests, and the impact on public health [18]. SARS-CoV-2 antigen testing is somewhat singular in this respect, as there are few definitive treatments widely available, and testing is crucial as a public health matter. Antibody testing is somewhat more complex, as it can indicate current infection and/or history of infection (and with the advent of SARS-CoV-2 vaccines, the immune response to vaccination) and thus provides additional yet potentially ambiguous information. 

A review of the literature suggests that to date, few studies have examined the effect of SARS-CoV-2 antibody testing on pandemic-related anxiety, with only one study including HCWs. Studies in mechanic industry workers and university students have reported an association between SARS-CoV-2 testing and decreased general pandemic and infection-related fear and anxiety [19,20]. One study that included Emergency Department personnel and assessed perceived changes in anxiety following a single serology test found that participants agreed that knowledge of COVID-19 serology status decreased their anxiety and that those who received a positive antibody test result were more likely to report decreased anxiety [21]. There are, however, no studies that measure changes in anxiety over time with readily available repeated antibody testing.

The objective of the current study was to examine the effects of repeated SARS-CoV-2 antibody testing on pandemic-related anxiety in HCWs.

## 2. Methods

### 2.1. Design and Participants

This longitudinal cohort study was conducted in two phases, from 14 April 2020 to 13 May 2020 and from 14 May 2020 to 13 July 2020, at a quaternary care children’s hospital. Participants (*n* = 362) included physicians, physician assistants (Pas), nurse practitioners (NPs), registered nurses (RNs), medical technicians, and secretaries/other administrative staff with direct patient contact or close proximity to patient-care areas in the Emergency Department (ED) (phase 1; *n* = 132) and in the Pediatric Intensive Care Unit (PICU), as well as with inter-hospital transport teams, ED teams, and the operative services teams (phase 2; *n* = 230).

### 2.2. Procedures

Individuals were recruited via department-wide emails (convenience sampling methodology). Informed consent was obtained from all participants. Participants completed a demographics questionnaire upon study enrollment and, prior to each serology test, rated their level of anxiety about the COVID-19 pandemic. Participant questionnaires were administered via a Research Electronic Data Capture tool (REDCap) [22]. Participants in phase 1 underwent serology testing every 96 h or prior to each ED shift. Participants in phase 2 underwent serology testing every seven days. All study procedures were approved by the internal Institutional Review Board. All methods were performed in accordance with the relevant guidelines and regulations.

### 2.3. Measures

Demographics: A demographics questionnaire assessed the HCWs’ self-reported age, gender, race, job position, and years of experience (post-training).

COVID-19 Pandemic Anxiety: HCWs’ current (i.e., state) anxiety was self-reported using a visual analogue scale (VAS). The measure instructed HCWs to indicate their “level of anxiety about the COVID-19 pandemic at this moment” along a 100 mm horizontal line with end points labeled “not anxious” (0) and “very anxious (100) [23]. The VAS is an established measure used to assess a variety of subjective states, including state anxiety in adults and healthcare workers, and is comparable to gold-standard, multi-item state anxiety measures such as the State-Trait Anxiety Inventory [23,24,25,26,27,28,29].

SARS-CoV-2 IgM and IgG antibody testing: Antibody levels were tested via the Wytcote Superbio SARS-CoV-2 IgM/IgG Antibody Fast Detection Kit (Colloidal Gold), which assessed for SARS-CoV-2 anti-spike IgM and IgG antibodies. The antibody kit received emergency use authorization from the Food and Drug Administration. The sensitivity of the assay is 95.2–100.0%, and the specificity of the assay is 98.1–98.5%.

### 2.4. Statistical Analysis

Descriptive statistics were used to characterize the study population. COVID-19 pandemic anxiety reported on test 1 was considered an individual’s baseline anxiety level, and changes in anxiety were calculated by subtracting this from subsequent responses (e.g., test 2 anxiety—baseline testing anxiety). Prior to conducting the primary study analyses, a repeated measures ANOVA indicated that there were no significant effects of study phase on changes in anxiety over time (i.e., no phase X time interaction; *F* = 0.47, *p* = 0.50). Further analyses of changes in anxiety therefore combined phase 1 and phase 2 participants. Repeated measures ANOVAs were used to examine changes in anxiety across testing time points and to assess whether demographic factors, study phase, and/or receiving a positive IgG or IgM test were associated with changes in anxiety. Given that the repeated measures pairwise comparisons indicated that anxiety significantly decreased from test 1 to test 3 (*p* < 0.034), with no changes in anxiety after test 3, subsequent analyses only examined the changes in anxiety from baseline to test 3. Changes in anxiety from baseline to test 3 were first dichotomized into two different variables. The first identified individuals that reported decreased anxiety versus those who did not. The second identified individuals that reported a clinically meaningful decrease in anxiety versus those who did not. A > 30% decrease in pandemic-related anxiety was used as a threshold for a clinically meaningful decrease in pandemic-related anxiety, which is based on previous research assessing clinically meaningful changes in VAS scores [30,31]. For both variables, bivariate mean difference and chi-squared analyses examined whether HCW demographics differed across groups that reported decreased anxiety versus those that did not. Variables significant in the bivariate analyses were entered into a logistic regression model to examine unique effects on the likelihood of reporting decreased anxiety. A power analysis conducted using G*power [32] indicated that a sample of 45 would yield 80% power to detect a small to moderate effect (*f* = 0.15) for a repeated measures analysis with seven measurements. Power analyses for three repeated measurements indicated that, assuming a small to moderate effect, a sample of 74 would yield 80% power to detect a significant effect at alpha 0.05. The threshold for statistical significance was *p* < 0.05. Statistical analyses were conducted using IBM SPSS Statistics for Windows, Version 27.0. Armonk, NY, USA: IBM Corp.

## 3. Results

A total of 362 individuals participated in this study. Data were available for 206 participants across all 7 consecutive testing time points, and 320 participants across the first 3 consecutive testing time points. A total of 31 (8.6%) individuals tested IgM+, 49 (13.5%) tested IgG+, and 55 (15.2%) were either IgM or IgG positive. Participant demographics are presented in Table 1.

### 3.1. Baseline Anxiety

Overall, baseline COVID-19 anxiety was significantly higher in HCWs with less than 10 years of experience (*Z* = −2.63, *p* = 0.009) as well as in females compared to males (*Z* = −3.66 *p* < 0.001). Nurses reported higher COVID-19 anxiety compared to physicians, PAs, and technicians (*F* (3,302) = 6.04, *p* = 0.003; Table 2). PICU HCWs reported significantly higher baseline anxiety than those in the ED and transport (*F* (3,301) = 4.34, *p* = 0.005; Table 2). When stratifying by department, the effect of gender on baseline anxiety was only significant in the ED, the effect of years of experience on baseline anxiety was only significant in the OR/Pre-op (*r*_s_ = −0.29, *p* = 0.002), and the effect of job position on baseline anxiety was not significant, though anxiety scores were higher for nurses in all groups.

### 3.2. Changes in Anxiety

Repeated measures analyses indicated that anxiety decreased after baseline in both positive and negative test participants; however, after the third test, reported anxiety in those with positive test results increased, whereas anxiety in the negative group remained stable and did not significantly change following test 3 (Figure 1, (*F*(5,1020) = 3.75, *p* = 0.002)). Among those who had a positive test, the type of test (i.e., IgM vs. IgG) did not have an effect on changes in anxiety scores (*F* (5,92) = 1.44, *p* = 0.22).

Of the HCWs who reported decreased anxiety from baseline to test 3, 29.8% (*n* = 57) had a clinically meaningful decrease in anxiety (i.e., ≥30% decrease). Demonstrating a clinically meaningful decrease in anxiety was disproportionately associated with being a HCW in the ED compared to a non-ED HCW (*X*^2^ = 5.16, *p* = 0.023). A subsequent logistic regression analysis examined the association between being an ED HCW vs. a non-ED HCW and likelihood of reporting a clinically meaningful decrease in anxiety. The regression results indicated that ED HCWs were 1.97 times [95% CI (1.09, 3.56)] more likely to report a clinically meaningful decrease in anxiety compared to non-ED HCWs (*X*^2^(1) = 5.05, *B* = 0.68, *SE* = 0.30, *p* = 0.025).

Demographic or occupational factors, including gender, age, race, years of experience, job position, and department, did not affect the likelihood of reported anxiety scores decreasing from baseline to test 3.

## 4. Discussion

This study shows that repeated SARS-CoV-2 antibody serology testing is associated with decreasing perceived anxiety levels in pediatric HCWs, suggesting that access to testing is an important mitigation factor for HCW COVID-19-related anxiety. Occupational variables including but not limited to access to personal protective equipment, COVID-19 patient volumes, and taking on additional clinical roles also play an important role in HCW anxiety; however, these were not able to be controlled for in this study. When healthcare systems aim to address pandemic-related anxiety and burnout, access to testing is only one of many facets to address.

Under the conditions of this study, we found that baseline reported anxiety regarding the COVID-19 pandemic was, in general, statistically significantly higher in women, nurses, and those with fewer years of post-training work experience. This is congruent with a meta-analysis of 65 studies examining the mental health of HCWs during the COVID-19 pandemic, in which Batra et al. found a higher prevalence of anxiety in both females compared to men and nurses compared to physicians [7]. It is notable, however, that the prevalence of diagnosed anxiety disorders in the general population is more common in women [33]. Interestingly, when stratifying our results by department, the significance of gender disappeared except for in the ED, and the effect of job position was also not statistically significant.

Elbay et al. investigated anxiety, stress, and depression in physicians during the COVID-19 pandemic and found that less work experience was associated with higher anxiety scores [34]. There are likely a multitude of reasons at the root of this finding, including another proposed source of anxiety during the COVID-19 pandemic, namely that competency in new/unfamiliar tasks or longer tenure in the field may be associated with a more robust personal resource base [12]. However, again, when stratifying by department, the effect of years of experience on baseline anxiety was only significant in one department (OR/Pre-op).

Although those in the PICU also reported higher levels of baseline anxiety, this is somewhat difficult to interpret as those working in the PICU only participated in phase 2, which was itself associated with higher levels of baseline anxiety regarding the COVID-19 pandemic. There are a potential myriad of reasons why baseline anxiety may have increased between phase 1 and phase 2, including differing study populations and rising case counts in the surrounding areas, with a community prevalence in phase 1 of 99.5 per 100,000 compared to 458 per 100,000 during phase 2 [35,36].

This study also demonstrated that from baseline to test day 3, anxiety decreased in participants who tested positive or negative. After test day 3, reported anxiety in those with positive results increased back towards baseline anxiety levels, whereas decreased anxiety levels in the negative group were maintained through test day 7. This appeared true for the majority of the study population; no factors were found that affected this trajectory. There are numerous potential rationales to explain this finding, including that previous negative results may have been equated to a lack of current infection and reassurance (irrespective of validity) that an individual’s precautions against infection had been effective. In contrast, Rodriguez et al. found that receiving a positive serology test was associated with decreased reported anxiety [21]. However, Rodriguez et al. measured anxiety levels 2–3 weeks after participants received their serology results [21]. Therefore, at that time, it would have been clear to individuals that the serology results indicated past infection (and potential partial protection from future infection).

Timing has been shown to be important with respect to testing and anxiety in studies of other infectious diseases as well. Campbell et al., for example, studied the effect of population testing for chlamydia on anxiety and found that testing reduced anxiety in both males and females, although anxiety was reduced in males at the time of testing whereas female anxiety was reduced upon receipt of negative results [15]. Interestingly, Campbell et al. found no increase in anxiety in those invited to test. Similarly, Eborall et al. found that invitation to test for type 2 diabetes did not increase anxiety, and Kaerlev et al. found that screening for lung cancer was not associated with an increase in the number of filled antidepressant and/or anxiolytic prescriptions [14,37]. This is reassuring, as one of the most significant limitations of this study is that we did not measure anxiety levels prior to the pandemic or prior to participant enrollment in this study. We were thus unable to discern if baseline testing anxiety and changes in anxiety were associated with HCWs’ pre-pandemic or pre-testing anxiety.

Other limitations included the measurement of reported anxiety/perceptions of anxiety levels only and not attempting to assess what “anxiety” meant to individuals participating in this study. Participants were not asked why they reported a decrease or increase in anxiety, and HCW anxiety related to the serology testing procedures and/or test results was not assessed and may have affected reported anxiety levels. An important confounder in this study is that those testing positive for antibody serology were unable to work pending negative PCR results (up to 36 h), the significance of which would depend on an individual subject’s perceived consequence of this result. Although the VAS is considered a valid and reliable measure for assessing state-dependent anxiety, the use of a one-item measure limits sensitivity and the assessment of different dimensions of anxiety [29]. In addition, although the VAS has been used to determine change in clinical levels of anxiety, participants in this study indicated their pandemic-related anxiety specifically; we did not include an assessment of clinical anxiety [30]. Future studies would benefit from the inclusion of a clinically validated multi-item measure of anxiety disorder symptoms, which may allow for a more in-depth analysis of clinical levels of anxiety and better inform psychological intervention. Finally, given the small proportion of participants that who positive serology test results, we may have been underpowered to detect significant effects within the positive participant sub-sample.

## 5. Conclusions

This study suggests that repeated SARS-CoV-2 antibody serology testing may be associated with decreasing anxiety levels. It is critically important to identify factors that may help protect healthcare worker mental health. If the results of this study are validated by further research, access to repeated testing may represent one strategy to attenuate pandemic-related anxiety in healthcare workers. There are many remaining questions, and the results of this study may help guide future research. Important additional questions include examining the number and frequency of tests that may confer the most benefit, and the utility of antigen testing or vaccine response (as many healthcare workers are now vaccinated) in decreasing anxiety. Similar studies might also attempt to identify the specifics driving this potential decrease in anxiety.

## Figures and Tables

**Figure 1 ijerph-19-09562-f001:**
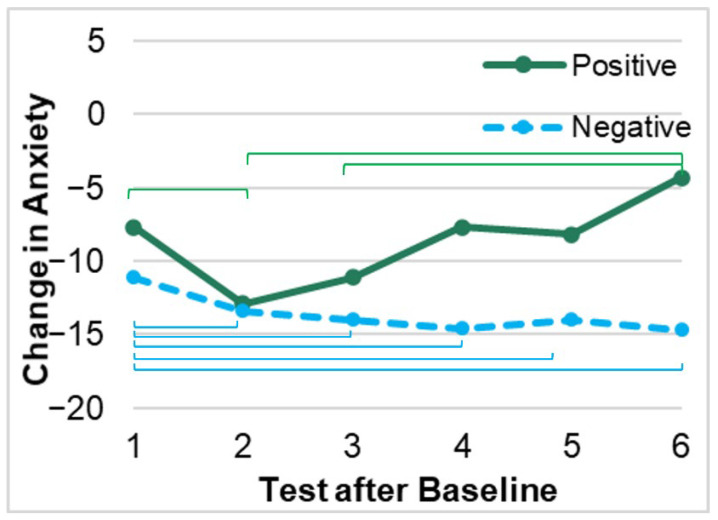
Changes in anxiety with the test results. Note: brackets indicate significant differences (*p* < 0.05) within each group.

**Table 1 ijerph-19-09562-t001:** Participant demographics.

	Negative Sample (*n* = 307)	Positive Sample *(n* = 55)
	Median (IQR)	Median (IQR)
Years of Experience	10.00 (13.00)	9.00 (14.00)
	*n* (%)	*n* (%)
Age		
≤30 years	98 (31.9)	20 (36.4)
31–40 years	98 (31.9)	17 (30.9)
41–50 years	72 (23.5)	13 (23.6)
≥51 years	39 (12.7)	5 (9.1)
Gender		
Female	219 (71.3)	38 (69.1)
Male	88 (28.7)	17 (30.9)
Race/Ethnicity		
African American, Black	5 (1.6)	0 (0)
American Indian/Alaskan	1 (0.3)	0 (0)
Asian, Pacific Islander	64 (20.8)	8 (14.5)
Hispanic/Latino	55 (17.9)	13 (23.6)
White	159 (51.8)	30 (54.5)
Multi-Racial	18 (5.9)	2 (3.6)
Other	5 (1.6)	2 (3.6)
Department		
Emergency Department	120 (39.1)	13 (23.6)
Operating Room Pre-Operative	91 (29.6)	17 (30.9)
Pediatric Intensive Care Unit	60 (19.5)	17 (30.9)
Transport	36 (11.7)	8 (14.5)
Position		
Physician	58 (18.9)	11 (20.0)
Physician Assistant	10 (3.3)	0 (0)
Nurse Practitioner, Registered Nurse	166 (54.1)	28 (50.9)
Tech or Administrative	73 (23.8)	16 (29.1)

Note. No significant differences between groups.

**Table 2 ijerph-19-09562-t002:** Baseline COVID anxiety across healthcare worker demographic groups.

	Baseline Anxiety Mean (SD)				*p*-Value
Gender	Female	Male			0.009
	51.55 (24.93) ^a^	39.56 (27.29) ^b^			
Job Position	Physician/ PA	Nurse	Technician		0.003
	44.70 (25.40) ^a^	52.84 (25.07) ^b^	40.69 (27.28) ^a^		
Department	Emergency Department	Intensive Care Unit	Peri-Operative	Transport	0.002
	43.42 (26.73) ^a^	55.75 (23.39) ^b^	49.61 (26.12) ^a,b^	45.00 (26.52) ^a^	

Note. Significant differences (*p* < 0.01) within each variable group are denoted by different superscripts; PA = physician assistant.

## Data Availability

The de-identified datasets generated during and/or analyzed during the current study are available from the corresponding author upon reasonable request.

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
