# Peer review of "Recurrent SARS-CoV-2 Serology Testing and Pandemic Anxiety: A Study of Pediatric Healthcare Workers"

_ijerph, 2022, doi:10.3390/ijerph19159562_

Round 1

Reviewer 1 Report

This is an interesting study on COVID-19. The study topic is novel and important.

However, the following changes should be considered to improve the quality of the manuscript:

1. Introduction: please provide a more comprehensive rationale for this study. Why this kind of study is important? What this kind of study may add. There are numerous COVID-related papers, so please emphasize the importance of this study.

2. Please clearly define the tool that was used to assess the anxiety level. Do the authors use validated and well-known tools? How about the interpretation of the results?

3. The Authors should provide basic data on the antibody test used in this study (sensitivity, specificity, FDA or EMA approval, etc.) These data are crucial to assess the methodology of this study.

4. The Authors should improve the results section. Please consider separate columns for different HCWs.

5. Moreover, the type of work (direct contact with patients; other types of work) should be considered.

6. Table 3 is unclear, please improve it.

7. Please add 2-3 sentences on practical implications of this study.

Author Response

Dear Editors and Reviewers,

Thank you very much for considering our manuscript, “Recurrent SARS-CoV-2 Serology Testing and Pandemic Anxiety: A Study of Pediatric Healthcare Workers”. Below, we have addressed each point raised by the reviewers. We thank the reviewers for their helpful comments and believe the revised manuscript is a stronger product as a result.

Thank you for your time and consideration. Please do not hesitate to contact me should you have any questions or concerns.

Sincerely,

Theodore Heyming, MD

Children’s Hospital of Orange County

505 S. Main Street, Suite 940 | Orange, CA 92868, USA

P (949)975-9247 [email protected]

  1. Introduction: please provide a more comprehensive rationale for this study. Why this kind of study is important? What this kind of study may add. There are numerous COVID-related papers, so please emphasize the importance of this study.

In this study we attempted to identify if repeat antibody testing may be an effective tool for reducing pandemic related anxiety. Healthcare workers undergo unique stress during pandemics, and it is important to explore possible interventions for now and for the future with respect to protecting healthcare worker mental health. We have clarified the goals of this study, and how it is unique in the revised introduction (lines 91-102).

  1. Please clearly define the tool that was used to assess the anxiety level. Do the authors use validated and well-known tools? How about the interpretation of the results?

In the methods section, we provided more information about the visual analog scale (VAS), including validity and well-established use of the VAS to measure anxiety (lines 112-115).

  1. The Authors should provide basic data on the antibody test used in this study (sensitivity, specificity, FDA or EMA approval, etc.) These data are crucial to assess the methodology of this study.

We added more information about the antibody test to the methods section, including the sensitivity and specificity.

  1. The Authors should improve the results section. Please consider separate columns for different HCWs.

We have made significant revisions to the results section to improve clarity and address the reviewers’ comments. We also reformatted Table 2 to more clearly present baseline anxiety data across different HCW demographic/occupational groups.  

  1. Moreover, the type of work (direct contact with patients; other types of work) should be considered.

Since we only included healthcare workers with direct patient contact or those working in patient care areas, we did not include patient contact in our analyses.

  1. Table 3 is unclear, please improve it.

We agree that Table 3 was unclear. We have removed this table.

  1. Please add 2-3 sentences on practical implications of this study.

Thank you for allowing us to address this. We have added practical implications of this study in the conclusion.

Reviewer 2 Report

This is an original research aimed at evaluate the relationship of repeated testing on COVID-19 related anxiety in pediatric frontline healthcare workers. Limitations of the use of a one-item  self-reported measure of anxiety should be recognized (in addition to those inherent to not evaluate pre levels of anxiety). 

Author Response

July 15, 2022

Dear Editors and Reviewers,

Thank you very much for considering our manuscript, “Recurrent SARS-CoV-2 Serology Testing and Pandemic Anxiety: A Study of Pediatric Healthcare Workers”. Below, we have addressed each point raised by the reviewers. We thank the reviewers for their helpful comments and believe the revised manuscript is a stronger product as a result.

Thank you for your time and consideration. Please do not hesitate to contact me should you have any questions or concerns.

Sincerely,

Theodore Heyming, MD

Children’s Hospital of Orange County

505 S. Main Street, Suite 940 | Orange, CA 92868, USA

P (949)975-9247 [email protected]

  1. Limitations of the use of a one-item self-reported measure of anxiety should be recognized (in addition to those inherent to not evaluate pre levels of anxiety).

We have added further discussion regarding the limitations of a one-item self-reported measure of anxiety to the limitations section of the discussion (lines 286-291).

Reviewer 3 Report

Introduction:

In this section, empirical evidence of the use of visual analog scales to assess anxiety, much less clinical anxiety, is not presented, since being one of the purposes of the study it should be substantiated.

The anxiety associated with the result of the serologic test may be different from the perception of anxiety before the COVID 19 and the anxiety perceived at the constant application of the antibody test.

It is suggested to review the objective of the study since the last section of the study is linked more to a research purpose that is not fulfilled.

Reflect that only the perception of the level of anxiety before the pandemic and the repeated tests was measured.

Clinical anxiety was not evaluated and this type of technique only measures subjective perception and the results show a clinical decrease in anxiety (there is no theoretical or empirical basis).

Method:

It is important to describe what type of sampling was used in the selection of the participants, and it is important to describe what RED Cap administration means.

Explain each of the phases of the research, because in Phase 1 it was taken every 96 hours and in Phase 2 every week (how many days).

Measuring instruments.

It is important to explain what was the instruction given to the participants to measure anxiety.

In the statistical analysis it is important to be more explicit by phase how the intra- and inter-phase analyses were performed.

Results:

Revise the layout of Table 1 and it is necessary to footnote what the abbreviations mean.

Table 2 is not clear what a and b mean in the table and when it is intended to read the table to know the significance.

Table 3: Reviewing the data in some indicators, the n and the % are inverted.

It is important to justify and explain how clinical anxiety was measured since it cannot be evaluated only with the visual analog scale.

They refer that they performed a logistic regression but there is no table indicating the indexes to interpret the results.

Discussion and conclusion

The authors refer in the background section and in the discussion how mental health is affected by the COVID 19 pandemic; however, the perception of anxiety is only one link of this construct. In addition, relating research findings to empirical evidence relates to occupational variables that are not assessed in this article.

It is important to mention the control of biases between the two phases evaluated and how these relate to the results.

The most important limitation is related to the assessment of anxiety since only the perception of anxiety in the face of the COVID19 pandemic was assessed and cannot be associated with clinical or nonclinical improvement of anxiety.

It is important to mention the sensitivity and specificity levels of the serological test used, although it has already been withdrawn from the market. 

Author Response

July 15, 2022

Dear Editors and Reviewers,

Thank you very much for considering our manuscript, “Recurrent SARS-CoV-2 Serology Testing and Pandemic Anxiety: A Study of Pediatric Healthcare Workers”. Below, we have addressed each point raised by the reviewers. We thank the reviewers for their helpful comments and believe the revised manuscript is a stronger product as a result.

Thank you for your time and consideration. Please do not hesitate to contact me should you have any questions or concerns.

Sincerely,

Theodore Heyming, MD

Children’s Hospital of Orange County

505 S. Main Street, Suite 940 | Orange, CA 92868, USA

P (949)975-9247 [email protected]

Introduction:

  1. In this section, empirical evidence of the use of visual analog scales to assess anxiety, much less clinical anxiety, is not presented, since being one of the purposes of the study it should be substantiated.

We acknowledge that our aim to assess COVID-related anxiety and not clinical anxiety was not clearly articulated. We have edited the introduction to clarify study objectives, which were to examine the effects of repeated antibody testing on pandemic-related anxiety in HCWs (lines 81-85). In the methods section, we edited the visual analogue scale description to include additional empirical evidence of the use of visual analogue scales to assess anxiety (lines 112-115).

  1. The anxiety associated with the result of the serologic test may be different from the perception of anxiety before the COVID 19 and the anxiety perceived at the constant application of the antibody test.

For the study anxiety assessment, participants were instructed to rate their anxiety about the COVID-19 pandemic, but we agree that anxiety related to the serology test result and/or testing procedures may have impacted participants’ report of anxiety about the pandemic. We have added this point to the limitations section (lines 274-278). We did not assess pre-COVID anxiety, which is also noted in the limitations section.

  1. It is suggested to review the objective of the study since the last section of the study is linked more to a research purpose that is not fulfilled.

We edited the end of the introduction to better articulate the study objective, which align with the study methods. 

  1. Reflect that only the perception of the level of anxiety before the pandemic and the repeated tests was measured.

We would like to clarify that we did not assess perception of pre-pandemic anxiety. As this study was conducted after the start of the pandemic, we were not able to measure level of anxiety before the pandemic, which is a noted limitation. We have clarified in the limitations section that we only measured perceptions of current anxiety.

  1. Clinical anxiety was not evaluated and this type of technique only measures subjective perception and the results show a clinical decrease in anxiety (there is no theoretical or empirical basis).

We acknowledge that the use of the term “clinically significant difference” may imply a change in clinical levels of anxiety, which, as the reviewer noted, we did not assess. In the revised manuscript, we use the term “clinically meaningful decrease” instead as this more closely aligns with terminology in the literature. In the statistical analyses section, we added details and citations to justify the use of a 30% decrease to represent a clinically meaningful change in pandemic-related anxiety. Specifically, we state, “A > 30% decrease in pandemic-related anxiety was used to as a threshold for a clinically meaningful decrease in pandemic-related anxiety, which is based on previous research assessing clinically meaningful change in VAS scores” (lines 141-144).

Method:

  1. It is important to describe what type of sampling was used in the selection of the participants, and it is important to describe what RED Cap administration means.

We have clarified that we used convenience sampling in the methods section. We have included what REDCap stands for and included a citation (line 101) regarding its use in research to the methods section.

  1. Explain each of the phases of the research, because in Phase 1 it was taken every 96 hours and in Phase 2 every week (how many days).

We edited the Procedures section to specify that “Participants in phase 2 underwent serology testing every seven days.” We changed the testing interval between Phase 1 and Phase 2 of the research study because of the low availability of testing kits during this time.

Measuring instruments.

  1. It is important to explain what was the instruction given to the participants to measure anxiety.

We added more details to the description of the VAS measure in the Methods, including that the “VAS measure instructed HCWs to indicate their “level of anxiety about the COVID-19 pandemic at this moment” along a 100-mm horizontal line with end points labeled “not anxious” (0) and “very anxious(100).”

  1. In the statistical analysis it is important to be more explicit by phase how the intra- and inter-phase analyses were performed.

Phase 1 and phase 2 data were combined for analyses given that preliminary analyses indicated that there was no significant effect of study phase on change in anxiety over time. This information is now added to the analyses section (lines 130-137).

Results:

  1. Revise the layout of Table 1 and it is necessary to footnote what the abbreviations mean.

We have deleted the abbreviations in all tables and written them out in the table, which eliminated the need for footnotes.

  1. Table 2 is not clear what a and b mean in the table and when it is intended to read the table to know the significance.

We reformatted this table to improve the presentation of the baseline anxiety results. Superscripts were used to indicate whether values within each variable group were significantly different (p<.01). Values with different superscripts were significantly different from one another (xa, xb); values with the same superscript were not significantly different (xa, xa).

  1. Table 3: Reviewing the data in some indicators, the n and the % are inverted.

We have removed Table 3. See response to Reviewer 1, comment #6.

  1. It is important to justify and explain how clinical anxiety was measured since it cannot be evaluated only with the visual analog scale.

Please see responses to comments #1 and #5. Clinical anxiety was not measured; self-reported pandemic-related anxiety was measured in this study.

  1. They refer that they performed a logistic regression but there is no table indicating the indexes to interpret the results.

We chose not to include a logistic regression table because the model only included one predictor variable. We edited this section of the results (lines 209-212) to include additional statistics that would typically be included in a logistic regression table (i.e., 95% CI of the odds ratio, unstandardized Beta weight of the predictor, and standard error). If the reviewer thinks this information would be more appropriate in a table, we can include a table in the second revision.

Discussion and conclusion

  1. The authors refer in the background section and in the discussion how mental health is affected by the COVID 19 pandemic; however, the perception of anxiety is only one link of this construct. In addition, relating research findings to empirical evidence relates to occupational variables that are not assessed in this article.

Thank you for this comment. We have added the following to our discussion, lines 222-227: “Occupational variables including, but not limited to, access to personal protective equipment, COVID-19 patient volumes, taking on additional clinical roles, also play an important role in HCW anxiety, and were not able to be controlled for in this study. When health care systems aim to address pandemic related anxiety and burnout, access to testing is only one of many facets to address.”

  1. It is important to mention the control of biases between the two phases evaluated and how these relate to the results.

Phase 1 and phase 2 data were combined for analyses given that preliminary analyses indicated that there were no significant effect of study phase on change in anxiety over time. This information is now added to the analyses section (lines 128-132).

  1. The most important limitation is related to the assessment of anxiety since only the perception of anxiety in the face of the COVID19 pandemic was assessed and cannot be associated with clinical or nonclinical improvement of anxiety.

Please see our response to comment #5 above regarding the use of a 30% decrease to represent a clinically meaningful change in pandemic-related anxiety. We acknowledge that current anxiety data do not reflect clinical anxiety and have addressed this in the limitations section of the discussion. Specifically, we state that “although the VAS has been used to demonstrate change in clinical levels of anxiety, participants in this study indicated their pandemic-related anxiety and we did not include an assessment of clinical anxiety.”

  1. It is important to mention the sensitivity and specificity levels of the serological test used, although it has already been withdrawn from the market. 

The test that was utilized was the Wytecote SARS-CoV-2 anti-spike IgM/IgG rapid test, which was approved under Emergency Use Authorization from the U.S. Food and Drug Administration. Per the package insert, the sensitivity of the assay is 95.2%-100.0% and the specificity of the assay is 98.1%-98.5%.

Reviewer 4 Report

The authors want to investigate baseline anxiety regarding the COVID-19 pandemic and changes in COVID-19 anxiety in conjunction with repeated SARS-CoV-2 antibody testing.

The manuscript title reflect the subject of the study.

The manuscript is well written, with proper use of scientific language. The introduction section has a good length. The methodology section is well written as well.

The results section is ok.

Kind Regards

Author Response

July 15, 2022

Dear Editors and Reviewers,

Thank you very much for considering our manuscript, “Recurrent SARS-CoV-2 Serology Testing and Pandemic Anxiety: A Study of Pediatric Healthcare Workers”. Below, we have addressed each point raised by the reviewers. We thank the reviewers for their helpful comments and believe the revised manuscript is a stronger product as a result.

Thank you for your time and consideration. Please do not hesitate to contact me should you have any questions or concerns.

Sincerely,

Theodore Heyming, MD

Children’s Hospital of Orange County

505 S. Main Street, Suite 940 | Orange, CA 92868, USA

P (949)975-9247 [email protected]

The authors want to investigate baseline anxiety regarding the COVID-19 pandemic and changes in COVID-19 anxiety in conjunction with repeated SARS-CoV-2 antibody testing.

The manuscript title reflect the subject of the study.

The manuscript is well written, with proper use of scientific language. The introduction section has a good length. The methodology section is well written as well.

The results section is ok.

            We thank the reviewer for these comments and believe the edits described above have improved the results section.

Round 2

Reviewer 1 Report

The Authors addressed all the comments.

Author Response

July 20, 2022

Dear Reviewer,

Thank you for your additional comments on our manuscript, “Recurrent SARS-CoV-2 Serology Testing and Pandemic Anxiety: A Study of Pediatric Healthcare Workers”. Below, we have addressed each point raised by the reviewers. We thank the reviewers for their helpful comments and believe the revised manuscript is a stronger product as a result.

Thank you for your time and consideration. Please do not hesitate to contact me should you have any questions or concerns.

Sincerely,

Theodore Heyming, MD

Children’s Hospital of Orange County

505 S. Main Street, Suite 940 | Orange, CA 92868, USA

P (949)975-9247 [email protected]

Reviewer 1:

  1. The Authors addressed all the comments.

We appreciate your careful review of our edits.

Reviewer 3 Report

Dear Authors, thank you very much for the effort to correct the article, I suggest that you review the abstract again and correct what you indicate about the clinical improvements in anxiety before the COVID19.

I suggest that future research should use measurement instruments with strict psychometric criteria in order to be unable to relate psychological variables that allow for more in-depth analysis and contribute to scientific knowledge in interdisciplinary studies.

Author Response

July 20, 2022

Dear Reviewers,

Thank you for your additional comments on our manuscript, “Recurrent SARS-CoV-2 Serology Testing and Pandemic Anxiety: A Study of Pediatric Healthcare Workers”. Below, we have addressed each point raised by the reviewers. We thank the reviewers for their helpful comments and believe the revised manuscript is a stronger product as a result.

Thank you for your time and consideration. Please do not hesitate to contact me should you have any questions or concerns.

Sincerely,

Theodore Heyming, MD

Children’s Hospital of Orange County

505 S. Main Street, Suite 940 | Orange, CA 92868, USA

P (949)975-9247 [email protected]

Reviewer 3:

  1. Dear Authors, thank you very much for the effort to correct the article, I suggest that you review the abstract again and correct what you indicate about the clinical improvements in anxiety before the COVID19.

We updated the methods section of our abstract to report a change in anxiety from the study baseline to attempt to correct what we indicate about the clinical improvements in anxiety.

  1. I suggest that future research should use measurement instruments with strict psychometric criteria in order to be unable to relate psychological variables that allow for more in-depth analysis and contribute to scientific knowledge in interdisciplinary studies.

We have added a sentence to our limitations section capturing this sentiment, which is on lines 295-299.